Identification and subcellular localization analysis of membrane protein Ycf 1 in the microsporidian Nosema bombycis

Chen Yong 1
Wei Erjun 1
Chen Ying 1
He Ping 1
Wang Runpeng 1
Wang Qiang 1 2
Tang Xudong 1 2
Zhang Yiling 1 2
Zhu Feng 3
Shen Zhongyuan szysri@163.com 1 2
1 School of Biotechnology, Jiangsu University of Science and Technology , Zhenjiang , Jiangsu , China
2 Chinese Academy of Agricultural Sciences, Institute of Sericulture , Zhenjiang , China
3 Zaozhuang University , Zaozhuang , Shangdong , China
Tang Xiaotian
Electronic publication date: 2022 Jul 8
Publication date: 2022
Volume: 10
Electronic Location ID: e13530
Received 2022 Jan 24; Accepted 2022 May 11
Copyright: ©2022 Chen et al.
Copyright year: 2022
Copyright holder: Chen et al.
License: This is an open access article distributed under the terms of the Creative Commons Attribution License, which permits unrestricted use, distribution, reproduction and adaptation in any medium and for any purpose provided that it is properly attributed. For attribution, the original author(s), title, publication source (PeerJ) and either DOI or URL of the article must be cited.
License URL: https://creativecommons.org/licenses/by/4.0/

Keywords: Microsporidia, Nosema bombycis, Membrane protein, Immunolocalization, RNAi

Funding: China Agriculture Research System of MOF and MARA National Natural Science Foundation of China 31960684 This work was funded by the China Agriculture Research System of MOF and MARA and the National Natural Science Foundation of China (31960684). The funders had no role in study design, data collection and analysis, decision to publish, or preparation of the manuscript.

==============================
Microsporidia are obligate intracellular parasites that can infect a wide range of vertebrates and invertebrates including humans and insects, such as silkworm and bees. The microsporidium Nosema bombycis can cause pebrine in Bombyx mori, which is the most destructive disease in the sericulture industry. Although membrane proteins are involved in a wide range of cellular functions and part of many important metabolic pathways, there are rare reports about the membrane proteins of microsporidia up to now. We screened a putative membrane protein Ycf 1 from the midgut transcriptome of the N. bombycis-infected silkworm. Gene cloning and bioinformatics analysis showed that the Ycf 1 gene contains a complete open reading frame (ORF) of 969 bp in length encoding a 322 amino acid polypeptide that has one signal peptide and one transmembrane domain. Indirect immunofluorescence results showed that Ycf 1 protein is distributed on the plasma membrane. Expression pattern analysis showed that the Ycf 1 gene expressed in all developmental stages of N. bombycis. Knockdown of the Ycf 1 gene by RNAi effectively inhibited the proliferation of N. bombycis. These results indicated that Ycf 1 is a membrane protein and plays an important role in the life cycle of N. bombycis.

Introduction

Microsporidia are obligate intracellular parasitic unicellular eukaryotes that can infect almost animal species including important agricultural species (Barandun et al., 2019; Franzen, 2008; Higes et al., 2008; Stentiford et al., 2016), as well as humans, especially immunocompromised patients (Corradi, 2015). N. bombycis, a parasite that can cause pebrine disease, is the first microsporidium discovered in the silkworm (Jie et al., 2017; Liu et al., 2013).

Membrane proteins play a crucial part of cell signaling, cellular energy production and the initiation of many cellular signaling cascades. Previous studies reported that membrane proteins can cooperate with a variety of accessory proteins, together which form biomolecular networks  that underlie the complex functions of living cells (Gupta, 2019; Wang et al., 2012). Many membrane-associated protein complexes, such as occludin (a cell adhesion molecule), take an important role in paracellular transport (Furuse, Sasaki & Tsukita, 1999; Li et al., 2021). As an obligate intracellular parasite, the microsporidia recognize the host cells through the proteins on the wall of spore, and infect the host by extruding the sporoplasm into the cell, during this process, the proteins on the polar tube and sporoplasm perform functions by interacting with the host (Peyretaillade et al., 2012). Due to partial loss of metabolic pathways, the microsporidia must depend on the nutrients provided by the host cells for its proliferation in the host cells. The transporter, such as ATP/ADP transporter and protein transporter, are the key for the microsporidia to obtain energy from the host (He et al., 2018; Heinz et al., 2014). Previous studies reported that the ATP/ADP transporter NoboABCG1.1 takes part in the infection process (He et al., 2018), and the sporoplasm surface protein 1 of Encephalitozoon hellem (EhSSP1) participates in energy acquisition by interacting with host cell mitochondrial outer membrane (Han et al., 2019).

In this study, we screened a putative membrane protein, Ycf 1, by using the midgut transcriptome of N. bombycis infected silkworm and the N. bombycis genome database (https://silkpathdb.swu.edu.cn/). The subcellular localization of Ycf 1 in the dormant spores and intracellular proliferative phase of N. bombycis were investigated by anti-Ycf 1 monoclonal antibody. The indirect immunofluorescence assay (IFA) results showed that the Ycf 1 was distributed on the plasma membrane of N. bombycis. β-tubulin is a housekeeping and conserved gene, its copy number can reflect the number of N. bombycis (Huang et al., 2018a). The copy number of Nbβ-tubulin was detected to investigate the role of Ycf 1 in the proliferation of N. bombycis. After knocking down the Ycf 1 gene by RNA interference, the copy number of Nbβ-tubulin was significantly reduced. These results indicated that Ycf 1 plays an important role in the life cycle of N. bombycis.

Materials & Methods

Parasite and host

The silkworm strain SuRong × Xiyu, BmN cell line, N. bombycis, Nb-actin polyclonal antibody and SWP30 polyclonal antibody were provided by the Sericultural Research Institute of Chinese Academy of Agricultural Sciences. The BmN cell was isolated from silkworm and kept in our laboratory.

Cloning of Ycf 1 gene and analysis

The purified spore suspension of N. bombycis (109 sopres/mL) was crushed in a bead grinder with 1:1 acid-washed glass beads for 1 min, cooled on ice for 5 min; this process was repeated six times. The genomic DNA (gDNA) of N. bombycis was extracted using fungal genomic DNA extraction kit (Sangong Bioengineering, Shanghai, China), and kept at −20 °C following concentration measurements.

The gene sequence of the putative membrane protein Ycf 1 (GenBank accession number:EOB13621.1) was searched from the NCBI database (https://www.ncbi.nlm.nih.gov) and the specific forward primer 5′-GCGCGGATCCATGAAATTTACTACTTTTTG-3′ (BamH I restriction site); reverse primer 5′-GCGCTCGAGTTATTTAGAAGCCATCAT-3′ (Xho I restriction site) were designed to amplify the target gene. The 50 µL PCR amplification reaction systems were as follows: PrimeSTAR HS DNA Polymerase (TaKaRa Biotechnology, Beijing, China) 25 µL, specific forward and reverse primers 2 µL (10 µM) respectively, gDNA 2 µL (100 ng), and ddH2O 19 µL . The PCR products were separated on a 1% agarose gel and extracted with an Axyprep DNAGel Extraction Kit, then cloned into the pMD19-T-Vector (TaKaRa Biotechnology, Beijing, China) with poly(A) tail. The pMD19-T-Ycf 1 vector was transformed into E.coli TOP10 competent cells (Sangong Bioengineering, Shanghai, China), and then cultured on LB plates containing ampicillin at 37 °C. The recombinants were identified by PCR and sequenced by Sangong Bioengineering (Shanghai, China).

The molecular weight and isoelectric point of Ycf 1 protein were predicted by Compute pI/Mw tool (https://web.expasy.org/compute_pi/). Signal peptide was predicted by SignalP-5 (https://services.healthtech.dtu.dk/service.php?SignalP-5.0). TMHMM Serverv.2.0 (https://services.healthtech.dtu.dk/service.php?TMHMM-2.0) was used to predict transmembrane domains. The phosphorylation and glycosylation sites were predicted by DTU Health Tech (http://cello.life.nctu.edu.tw/) and its subcellular location was predicted by CELLO v.2.5 (http://cello.life.nctu.edu.tw/).

Expression of recombinant protein of Ycf 1

The correctly sequenced pMD19-T-Ycf 1 positive plasmid and pET-28a empty plasmid were identified by double enzyme digestion. The pET-28a-Ycf 1 was transformed into E.coli BL21 Star (DE3) (Sangong Bioengineering, Shanghai, China). The correctly identified positive colonies were used for prokaryotic expression.

The overnight cultured bacterial cells were added to LB liquid medium (Kanamycin, 100 mg/mL) at ratio of 1:100, then incubated at 37 °C in a shaker until the OD600 reached 0.6. One mL of the bacterial cells were collected as negative control before induction. The remaining bacterial cells were induced with different isopropyl β-D-thiogalactoside (IPTG) concentrations for 20 h at 20 °C. Ultrasonic disruption (250 W, 3 s, 5 s interval) was performed for about 10 min, the supernatant and precipitate were collected separately by centrifugation, and boiled with 1 × protein loading buffer for 10 min at 100 °C, respectively. SDS-PAGE was performed to confirm the expression of the recombinant protein, and the recombinant protein was tested by Western blot using Anti-His Tag antibody.

Preparation of Ycf 1 monoclonal antibody and Western blot analysis

The monoclonal antibody of Ycf 1 was prepared by Abmart Biopharmaceuticals in Shanghai, China referring to the method of previous report (Zhang et al., 2007).

The total protein of N.bombycis was extracted by the acid-washed glass beads (Dai et al., 2019). Western blot analysis was performed using Ycf 1 monoclonal antibody (overnight, 4 °C) as the first antibody and an HRP-conjugated goat anti-mouse IgG antibody (Sangong Bioengineering, Shanghai, China) as the second antibody (room temperature, 1 h). The mouse IgG was used as negative control. The PVDF membrane was incubated with Tanon™ High-sig ECL Western blot substrate (2 min), imaged with a Tanon 5200 Multi imaging system.

Immunolocalization of Ycf 1in N. bombycis

Dormant mature spore suspension was added to poly-lysine coated coverslips, airdried and fixed with 4% paraformaldehyde for 30 min. The monoclonal antibody of Ycf 1 (0.93 mg/mL) and the polyclonal antibody of spore wall protein of N. bombycis SWP30 (2.021 mg/mL) were visualized with Alexa Fluor 488 labeled Goat anti-mouse IgG (0.5 mg/mL, Sangong Bioengineering, Shanghai, China) or Cy5 labeled Goat anti-Rabbit IgG (0.5 mg/mL, Sangong Bioengineering, Shanghai, China), respectively. The immunolocalization was observed under a fluorescence inverted microscope (Olympus IX-71). The detailed steps of immunolocalization referred to the method of Qi et al. (2021).

The subcellular localization of Ycf 1 in the developmental stage of N. bombycis in BmN cells was investigated as the following procedure. The spore suspension of N. bombycis and 0.2 M KOH solution were pre-heated for 2 h at 27 °C, respectively, then mixed at the ratio of 1:1 (volume) and incubated for 40 min at 27 °C. The germinated spore suspensions were slowly dropped into flasks containing BmN cells at ratio of 10:1. After incubating for 1 h, the culture medium free of serum was replaced with TC-100 insect culture medium containing 10% serum (Gibco, Thermo Fisher Scientific, Australia), which was replaced with fresh culture medium after cultivation overnight. The BmN cell infected with N. bombycis at different days were added into the 6-well plate containing coverslips at different days post-infection. After washing three times with PBST, the infected BmN cell was fixed with 4% paraformaldehyde for 30 min. The next procedure is the same as IFA for dormant mature spores.

RNAi

Newly molted fifth instar silkworm larvae were fed on mulberry leaves for 6 h which were smeared with spores of N. bombycis (108 spores/mL). 3 µL of siRNA (1 µg) and a nonsense fragment as the negative control were injected into the infected silkworm, while ddH2O was used as blank control. The siRNA (Table 1) was synthesized by Sangong Bioengineering in Shanghai, China. The midgut tissues of the silkworm were collected at 24 h, 48 h, 72 h, 96 h, and 120 h post-infection, respectively, then washed with PBS and stored at −80 °C.

The midgut tissue was lysed with one mL of RNAiso plus lysate (TaKaRa Biotechnology, Beijing, China), then total RNA was extracted with Mini BEST Universal RNA Extraction Kit (TaKaRa Biotechnology, Beijing, China). The cDNA was synthesized with PrimerScript® RT Master Mix (TaKaRa Biotechnology, Beijing, China). The qPCR was performed according to the TB GreenTM Premix ExTaqTM II (Tli RNase H Plus) kit (TaKaRa Biotechnology, Beijing, China) manufacturer’s instructions with the primers as shown in Table 1. The β-tubulin gene of N. bombycis was served as reference gene. The transcriptional levels were calculated by the 2−ΔΔct values method with three biological replicates. The multiple t tests were conducted by using GraphPad Prism 8.0 (GraphPad Software, San Diego, CA, United States).

Table 1 Primer sequences for RNAi and qRT-PCR.

Used	Sense/ Antisense	Sequences	
Ycf 1 (RNA Oligo)	Sense	GCUGGGUGGUUUAGGAUCATT (5′-3′)	
	Antisense	UGAUCCUAAACCACCCAGCTT (5′-3′)	
NC (RNA Oligo)	Sense	UUCUCCGAACGUGUCACGUTT (5′-3′)	
	Antisense	ACGUGACACGUUCGGAGAATT (5′-3′)	
Ycf 1 (qRT-PCR)	Forward	GACCCGAACCTCCTGTTAAAGACC (5′-3′)	
	Reverse	CATAGCCACAACAACAACGAATCCG (5′-3′)	
β-tubulin (qRT-PCR)	Forward	TTCCCTTCCCTAGACTTCACTTC (5′-3′)	
	Reverse	CAGCAGCCACAGTCAAATACC (5′-3′)	
Nbβ-tubulin-qF(R)	Forward	AGAACCAGGAACAATGGACG(5′-3′)	
	Reverse	AGCCCAATTATTACCAGCACC(5′-3′)	

In order to analyze the effect of knockdown of Ycf 1 gene on the proliferation of N. bombycis, specific primers Nb β-tubulin-qF and Nb β-tubulin-qR (Table 1) were designed to detect the copy number of Nbβ-tubulin. The standard template was prepared according to the method of previous report (Huang et al., 2018a), and the standard curve covered five orders of magnitude (5.6 × 102–106). The multiple t tests were conducted by using GraphPad Prism 8.0 with three biological replicates.

Results

Cloning and expression of Ycf 1and immunoblot analysis

The Ycf 1 gene contains a complete ORF of 969 bp that encodes a 322 amino acid polypeptide. PCR amplification and sequencing results showed that the similarity is 99.7% compared to the sequence of N. bombycis CQ1 in the NCBI database, there are two base differences (291 T-C 301 G-A), and one amino acid difference (100 E-K). Ycf 1 has one signal peptide and one transmembrane domain with a pI of 6.49. Secondary structure analysis showed that the alpha-helix, random coils, beta sheets and extended fragment accounts for 32.3%, 57.14%, 4.35% and 6.21%, respectively. The phosphorylation and glycosylation prediction showed that it has thirty-six phosphorylation sites and one O-glycosylation site.

The double digestion of recombinant plasmid resulted in a band corresponding to the target gene (Fig. 1A). Western blot showed a target band about 50 kDa that was larger than the predicted size, presumably due to phosphorylation or glycosylation modification. The recombinant protein was detected in the precipitate after sonication, indicating that the protein was expressed as inclusion body in the E. coli BL21 DE3 (Fig. 1B). The optimum induction conditions were 0.5 mM IPTG for 20 h at 20 °C.

Figure 1 Double digestion and Western blot analysis of recombinate Ycf 1.

(A) Identification of recombinant plasmids by double digestion. Lane M: DL5000 DNA Ladder marker. Lane 1: pET-28a-Ycf 1 recombinant plasmid (arrow is target Ycf 1 gene). Lane 2: pET-28a vector. (B) Western blot identification of Ycf 1 prokaryotic expression. Lane M: protein molecular weight marker. Lane 1: Recombinant bacteria without induction . Lane 2: The supernatant of recombinant bacteria induced with 0.3 mM IPTG. Lane 3: The precipitate of recombinant bacteria induced with 0.3 mM IPTG. Lane 4: The supernatant of recombinant bacteria induced with 0.5 mM IPTG. Lane 5: The precipitate of recombinant bacteria induced with 0.5 mM IPTG. Lane 6: The supernatant of recombinant bacteria induced with 0.7 mM IPTG. Lane 7: The precipitate of recombinant bacteria induced with 0.7 mM IPTG.

Specific detection of monoclonal antibody

Western blot analysis showed that specific bands were detected both in the recombinant protein and extracted protein of N. bombycis while the mouse IgG was used as negative control (Fig. 2). These results indicated that the prepared monoclonal antibody has a specific antigen-antibody reaction and can be used in subsequent experiments.

Figure 2 Western blot identification of Ycf 1 protein in N. bombycis.

(A) Specificity analysis of Ycf 1 antibody. Lane M: protein molecular weight marker. Lane 1: Recombinant bacteria without induction. Lane 2: Recombinant bacteria induced with 0.7 mM IPTG. Lane 3: The precipitate of recombinant bacteria induced with 0.7 mM IPTG. Lane 4: Extracted protein of N. bombycis. (B) mouse IgG control. Lane 1: Recombinant bacteria without induction. Lane 2: Recombinant bacteria induced with 0.7 mM IPTG. Lane 3: The precipitate of recombinant bacteria induced with 0.7 mM IPTG. Lane 4: Extracted protein of N. bombycis.

Co-localization of Ycf 1 with SWP30 or Nb-actin in dormant spores of N. bombycis

In order to explore whether Ycf 1 is located on the plasma membrane in the mature spores of N. bombycis, we performed a co-localization analysis of Ycf 1 with SWP30 or Nb-actin. The Ycf 1 antibody and SWP30 or Nb-actin antibody were labeled with Alexa Fluor 488 (green) or Cy5 (red), respectively, while the nucleus was stained with DAPI (blue). SWP30 is a spore wall protein of N. bombycis and located on the endospore of mature spores (Wu et al., 2010). The IFA results showed that the green fluorescence was distributed on the inner side of mature spores, very close to the red fluorescence but not completely overlapped (Fig. 3A), the negative control has no green fluorescence (Fig. 3B). Nb-actin is a kind of multifunctional and indispensable protein of the cytoskeleton in the N. bombycis (Kühn & Mannherz, 2016). Co-localization results showed that the Ycf 1 protein was distributed on the inner side of the mature spore wall whereas the Nb-actin was distributed throughout the whole mature spore (Fig. 3C), the negative control has no green fluorescence (Fig. 3D). These results indicated that Ycf 1 protein is distributed on the plasma membrane of N. bombycis.

Subcellular localization of Ycf 1 in proliferative phase of N. bombycis

To further determine whether Ycf 1 protein is on the plasma membrane in the proliferative phases of N. bombycis, IFA was performed to investigate the subcellular localizations both individual Ycf 1 or with Nb-actin as reference in proliferative phase of N. bombycis. During infection, the N. bombycis invaded the host by injecting sporoplasm into the host cells. The Ycf 1 protein was mainly distributed on the plasma membrane of sporoplasm while the Nb-actin was distributed in the whole sporoplasm (Figs. 4A, 5A). Entering the proliferative phase, the meront is long fusiform and has multiple nuclei, Ycf 1 was distributed in the periphery, especially on the ends and connecting part of the dividing cells, whereas the Nb-actin appeared throughout the whole meront (Figs. 4B and 5B). In the sporogonic phase, the sporont has four nuclei, one sporont forms two sporoblasts through cytokinesis. The Nb-actin was present throughout the cell while Ycf 1 was mainly localized around the plasma membrane (Figs. 4C, 4D, 5C and 5D). There is no green fluorescence in the negative control (Figs. 4E and 5E). The above results further illustrated that Ycf 1 is a membrane protein of N. bombycis.

Figure 3 Co-localization of Ycf 1 with SWP30 or Nb-actin in dormant spores of N. bombycis.

(A) Dormant spore; white arrow: Ycf 1; yellow arrow: SWP30. (B) Mouse IgG. (C) Dormant spore; white arrow: Ycf 1; yellow arrow: Nb-actin. (D) Mouse IgG; Scale bars, 5 µm.

Figure 4 Subcellular localization of Ycf 1 protein.

Ycf 1 antibody coupled with Alexa Fluor 488 (green) labeled secondary antibody. DAPI (blue) was used to stain the nuclei of host cells and N. bombycis. (A) Sporoplasm. (B) Meront. (C) Sporont. (D) Sporoblasts. (E) Mouse IgG; red arrow: Ycf 1; Scale bars, 5 µm.

Figure 5 Colocalization of Ycf 1 with Nb-actin during the proliferative phase of N. bombycis.

Ycf 1 antibody coupled with Alexa Fluor 488 (green) labeled secondary antibody. Nb-actin antibody coupled with Cy5 (red) labeled secondary antibody. DAPI (blue) was used to stain the nuclei of host cells and N. bombycis. (A) Sporoplasm. (B) Meront. (C) Sporont. (D) Sporoblasts. (E) Mouse IgG; white arrow: Ycf 1; yellow arrow: Nb-actin; Scale bars, 5 µm.

Knockdown of Ycf 1 gene inhibited the proliferation of N. bombycis

For understanding the interference effect on Ycf 1 gene by RNAi, the expression level of Ycf 1 gene was detected by qRT-PCR using the β-tubulin of N. bombycis as reference gene. The relative expression of Ycf 1 gene decreased remarkably at 48 h compared to 24 h, then reached the highest at 72 h in the control group (Fig. 6A, Fig. S1B). Expectedly, RNA interference down-regulated the transcriptional level of the Ycf 1 gene (Fig. 6A, Fig. S1A). The interference effect on Ycf 1 gene was significant at 24 h and 72 h ( P < 0.05) and extremely significant at 96 h ( P < 0.01) (Fig. 6A). After knockdown of the Ycf 1 gene, the copy number of Nbβ-tubulin was investigated to reflect the effect on proliferation of N. bombycis. It was shown that the copy number of Nbβ-tubulin increased gradually with the proceeding of infection time, reached the highest at 120 h both in the control group and RNAi group (Fig. S2). Compared to the control group, the copy number of Nb β-tubulin was reduced significantly at 24 h and extremely significantly at 72 h and 96 h (Fig. 6B). These results suggested that silencing of the Ycf 1 gene inhibited the proliferation of N. bombycis.

Figure 6 Effect of downregulated Ycf 1 on N. bombycis proliferation via RNAi.

(A) Effect of RNAi on Ycf 1 gene expression. (B) Effect of knockdowning Ycf 1 gene on copy number of Nbβ-tubulin. Error bars represent the standard deviations of three independent replicates (n = 3, mean ± SE, One asterisk (*) indicates p < 0.05, two asterisks (**) indicate p < 0.01).

Discussion

More than half of the proteins in cells can bind to membranes in different forms and have an influence in material transport, signal recognition, and enzyme catalysis. Well-known membrane proteins including sodium-coupled transporters and ATP synthases (White, 2009), especially the G protein-coupled receptors (GPCRs) are target for many developing drugs (Lagerström & Schiöth, 2008). Membrane proteins are also involved in the infection process of some parasites: During malaria infection, erythrocyte membrane proteins play an important part in host cell infection and intracellular parasite development (Castro, Mendez & Moneriz, 2021). The ER membrane protein complex facilitates infection of the single-stranded RNA viruses Zika and Dengue viruses (Lin et al., 2019). In the process of invading host cells, apicomplexan parasites mediate motility and infectivity by releasing transmembrane proteins to the surface of host cells (Santos et al., 2011). Membrane protein TCHTE is also involved in heme transport in the parasite Trypanosoma cruzi (Pagura et al., 2020)

The life cycle of microsporidia is divided into three stages: infective phase, proliferative phase and sporogonic phase (Guo et al., 2016; Huang et al., 2018b). The spore wall is the earliest and direct contact with the host, and the spore wall proteins of microsporidia play a significant role in the process of infection. There are multiple wall proteins, including EcSWP1, NbSWP3, NbSWP5, NbSWP26, have been identified in the microsporidia (Li et al., 2009; Li et al., 2012; Meng et al., 2014). Wang et al. predicted 83 potential spore wall proteins in the N. bombycis by genome-wide scanning (Wang et al., 2020). Previous research pointed out that the proteins on the plasma membrane have dramatic effects on host responses for parasite (Weidner & Overstreet, 2021).

Microsporidia evade the host’s defenses and invade the host by injecting sporoplasm into the cell. Some microsporidia such as Encephalitozoon hellem form a parasitophorous vacuole in the host, whereas others such as N. bombycis is in direct contact with the host cytoplasm. Although microsporidia still have an intact glycolytic pathway (Huang et al., 2018b; Wiredu et al., 2017), they have lost many genes for metabolic pathways (Nakjang et al., 2013; Pombert et al., 2012), including oxidative phosphorylation, β-oxidation of fatty acids, and the tricarboxylic acid cycle (Chen et al., 2013; Corradi et al., 2010; Pombert et al., 2012). So microsporidia must acquire energy from the host. Research showed that microsporidia manipulate host cell metabolism and cell biological processes to promote nucleotide synthesis and maximize ATP and nucleotide import potential (Dean, Hirt & Embley, 2016). The sporoplasm get the substances needed for proliferation by inhibiting its carbon metabolism and through the transporter on the surface of the plasma membrane (He et al., 2020). Knockdown of NoBoABCG1-1 gene of N. bombycis membrane protein by RNA interference can inhibit the proliferation of N. bombycis (He et al., 2018). Transmission electron microscopy showed that the interaction of EhSSP1 with VDAC probably plays an important part in energy acquisition by Encephalitozoon hellem via its role in the association of parasitophorous vacuole with the host mitochondria (Han et al., 2019). Genomic studies also suggest that with expansion of transporter gene families, more and more transporters such as nucleotide transporter (NTT) proteins, which can compensate for reduced metabolic capacity, have been discovered (Dean et al., 2018; Heinz et al., 2014; Sirintra et al., 2013). NTT is a key strategy adopted by microsporidia to exploit host cells (Heinz et al., 2014). Zheng et al. (2021) reported that the membrane protein of N. bombycis NbTMP1 may get involved in the proliferation of N. bombycis.

In the present study we identified a novel membrane protein Ycf 1 in the N. bombycis. The Ycf 1 protein has a transmembrane domain and is distributed on the plasma membrane of N. bombycis in all developmental stages.

The expression level of Ycf 1 gene is the highest at 72 h post infection which is consistent with that the N. bombycis proliferates rapidly at the proliferative phase (Iwano & Ishihara, 1991). After silencing of the Ycf 1 gene by RNA interference, the expression level of Ycf 1 gene was significantly decreased at 24 h, 72 h, and 96 h post infection. The copy number of Nbβ-tubulin was significantly reduced at 24 h, 72 h, and 96 h in the interference group, indicating that the Ycf 1 probably acts as an important factor in the proliferation of N. bombycis. Membrane transporters can affect a variety of endogenous and xenobiotic processes as well, which are involved in drug absorption, metabolism and can be used to instruct drug design (Keogh, 2012). Chemical drugs such as fumagillin and albendazole are effective in controlling microsporidia in bees and grasshoppers (Guruceaga et al., 2020; Johny et al., 2009). In Plasmodium falciparum, the PfABCG protein located on the plasma membrane is involved in lipid transport (Edaye & Georges, 2015; Tran et al., 2014). Nevertheless, whether the Ycf 1 has the function of substance transport and is the potential drug target need to be further studied.

Conclusions

In this study, we identified a membrane protein Ycf 1 in the N. bombycis for the first time. The Ycf 1 is distributed on the plasma membrane of N. bombycis throughout its life cycle. Knockdown of the Ycf 1 gene significantly inhibited the proliferation of N. bombycis, indicating that Ycf 1 is involved in the proliferation of N. bombycis and may be used as a potential target for prevention of pebrine disease. Further research is needed for the mechanism of interaction between the Ycf 1 and proteins of the host.

Supplemental Information

Supplemental Information 1 Dataset of standard curve line

Click here for additional data file.

Supplemental Information 2 Figure 6A and Figure S1 dataset

Click here for additional data file.

Supplemental Information 3 Figure 6B and Figure S2 dataset

Click here for additional data file.

Supplemental Information 4 Full length image of gels

Click here for additional data file.

Figure S1 Relative expression of Ycf 1 gene

(A) relative expression of Ycf 1 gene in RNAi group. (B) relative expression of Ycf 1 gene in control group. Error bars represent the standard deviations of three independent replicates. (One-way ANOVA analysis by 24h as control. n = 3, * p < 0.05, ** p < 0.01).

Click here for additional data file.

Figure S2 Copy number of Nbβ-tubulin after RNAi on Ycf 1.

(A) Copy number of Nb β-tubulin in RNAi group. (B) Copy number of Nb β-tubulin in control group. Error bars represent the standard deviations of three independent replicates. (One-way ANOVA analysis by 24h as control. n = 3, ** p < 0.01).

Click here for additional data file.

Supplemental Information 7 Statistical Reporting

Click here for additional data file.

Supplemental Information 8 Ycf 1 sequencing results

Click here for additional data file.

We are grateful to all who provided the means for us to access free software, which we have used and cited in this article. We would also like to thank all partners and laboratory staff for kindly help and criticism.

Additional Information and Declarations

Competing Interests

Author Contributions

DNA Deposition

Data Availability

The authors declare there are no competing interests.

Yong Chen conceived and designed the experiments, performed the experiments, authored or reviewed drafts of the article, and approved the final draft.

Erjun Wei performed the experiments, prepared figures and/or tables, and approved the final draft.

Ying Chen performed the experiments, prepared figures and/or tables, and approved the final draft.

Ping He performed the experiments, prepared figures and/or tables, and approved the final draft.

Runpeng Wang performed the experiments, prepared figures and/or tables, and approved the final draft.

Qiang Wang analyzed the data, prepared figures and/or tables, and approved the final draft.

Xudong Tang analyzed the data, prepared figures and/or tables, and approved the final draft.

Yiling Zhang analyzed the data, prepared figures and/or tables, and approved the final draft.

Feng Zhu analyzed the data, prepared figures and/or tables, and approved the final draft.

Zhongyuan Shen conceived and designed the experiments, authored or reviewed drafts of the article, and approved the final draft.

The following information was supplied regarding the deposition of DNA sequences:

The Ycf 1 sequences are available at NCBI GenBank: EOB13621.1.

https://www.ncbi.nlm.nih.gov/protein/EOB13621.1/

The following information was supplied regarding data availability:

The raw data are available in the Supplemental Files.

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
