# Peer review of "Identification and subcellular localization analysis of membrane protein Ycf 1 in the microsporidian Nosema bombycis"

_PeerJ, doi:10.7717/peerj.13530_

## Round 0.1 · original submission · Major Revisions

Dear authors,

Your article has been reviewed by 4 peer reviewers. The reviewers have provided evaluations and made recommendations for revisions to your manuscript. All the reviewers have raised some concerns about your interpretations and English language, which need to be carefully considered. Two reviewers also have a few concerns about the experimental design.

I invite you to respond to the reviewers' detailed comments and revise your manuscript. All the reviewers' comments need to be addressed before the manuscript can be accepted.

Thank you for submitting your manuscript to PeerJ and I look forward to receiving your revision.

Best,

Xiaotian Tang, PhD
Academic Editor, PeerJ
xiaotian.tang@yale.edu

Reviewer 1 ·

Basic reporting

The authors cloned and expressed a transmembrane protein Ycf1 from Nosema bombycis, and prepared monoclonal antibody, investigated its localization using the antibody, and found that knocking-down Ycf1 could inhibits proliferation of N.bombycis, suggest that Ycf1 is an essential gene in N.bombycis, and could be a target for development of anti-N.bombycis drugs. They studied the location and function of Ycf1, it is important to understand N.bombycis infection in silkworm. The article was well-written, and figures and tables were well prepared. But the English needs to be improved by a proficient English speaker or a professional editing service. And the format of references is not correct and uniform, some author names are not correct.

Experimental design

The study is within Aims and Scope of the journal, their hypothesis is clear, and confirmed the hypothesis using proper research methods such as monoclonal antibody, indirect fluorescence assay, and RNAi, the methods were described with sufficient detail.

Validity of the findings

they provided enough data/result to conclude that Ycf1 is an important membrane protein in proliferation of N.bombycis, confirmed their hypothesis.

Additional comments

some references are not correct, such as in line 306-308, 316-319, 345, 432-441, 448-450
in line 80, the concentration of the primer may be not correct
in line 99, the antibiotics should be written in full name, and the concentration is not clear
in 124-127, the sentences should be rewritten. The antibody could be labeled with a dye but not another antibody.
in 168-170, the sentences should be rewritten

Reviewer 2 ·

Basic reporting

In this work, the authors screened a putative membrane protein Ycf 1 from the N. bombycis. However, this manuscript should be further improved before ready for publication.

Major revisions

1) The Introduction chapter is composed of one big paragraph, which logic and contexts are not clear enough, and should be revised to make it more clear.

2) Figure 1 is the exactly the same as the prediction of transmembrane helices by the TMHMM Serverv.2.0, which screenshot is not appropriate to be directly used as a result.

3) Sections 3.3 and 3.4 could be integrated, because both sections describe the same thing and show quite similar results.

4) lines 281-283: Zheng et al. did not report that the NbTMP1 plays an role in spore germination, but just speculated that it may be involved in proliferation in their discussion.


Minor revisions

1) The “N.bombycis” should be “N. bombycis”.

2) The “ml” should be “mL”.

3) Line81: “ddH2O” is not correct.

4) Please used 24h, 72h, 96h and 24hpi, 72hpi, 96hpi properly. The two expressions have completely different meanings.

5) Species names in references should be in italic.

Experimental design

1) Lines 155-157, the BmGAPDH is not an appropriate reference to determine the function of Ycf1 in N. bombycis proliferation.

2) Figure 4, the subcellular localization of Ycf 1 was just detected by using the Ycf 1 antibody and DAPI, which are not sufficient to indicate that Ycf 1 locates on plasma membrane. This could be improved by using cell membrane dye.

Validity of the findings

1) Section 3.6, why was the expression of N. bombycis β-tubulin highest at 24 hpi, but lower at 48, 96 and 120 hpi? How to confirm that the silkworm midguts were infected?

Additional comments

no comment

Reviewer 3 ·

Basic reporting

Chen et al. identified one putative membrane protein Ycf1 in Nosema bombycis and IFA and RNAi results suggested that this protein may involve in the proliferation of N. bombycis in silkworm cells. Here are my comments/suggestions.
What is the full name of Ycf 1? It seems the sequence of Ycf 1 is very similar to CQ1 (2 nt and 1 aa difference). Did the function of CQ1 have been studied in N.bombycis or other related microsporidia? If yes, what is the function of CQ1? It is better to provide this information in the introduction.
In the discussion, the authors should mention how Ycf 1 can be used to prevent the pebrine disease in sericulture.
The manuscript is very difficult to read and there are many language and grammar errors.
Some figures are not necessary or could be combined. Fig.2 could be deleted. Fig.7 and Fig.8 should be combined.

Experimental design

In the RNAi experiment, information about how to synthesize siRNAs and the concentration of siRNAs used for injection needs to be provided.

Validity of the findings

It would be better if the authors could provide the count number of N.bombycis spores in midguts of silkworm after knockdown of Ycf 1 gene.

Additional comments

n/s

Reviewer 4 ·

Basic reporting

no comment

Experimental design

no comment

Validity of the findings

The "merge" in row B of Figure 4 does not seem to result from objective superposition. Moreover, besides the two ends of the meront marked by the authors, there are many other places where you can see green fluorescent signals. All of these may significantly reduce the credibility of the data. A more convincing result is recommended (transmission electron microscopy observations can be one of the alternatives).

The authors are invited to explain and discuss why RNAi had a significant effect at 96 h, while the spore proliferation was not inhibited (Figure 7), and why there was no significant effect of RNAi at 120 h but spore proliferation was significantly inhibited (Figure 8).

Additional comments

The manuscript "Identification and subcellular localization analysis of membrane protein Ycf1 in the microsporidian Nosema bombycis" by Chen et al., identified a membrane protein Ycf 1 from microsporidia N. bombycis, found Ycf1 was distributed on the plasma membrane of spores throughout the life cycle of N. bombycis. The authors further found that the proliferation of N. bombycis was significantly inhibited after silencing the expression of Ycf1 by RNAi technology, and they concluded that Ycf1 was involved in the proliferation of N. bombycis. However, the reliability of several key experimental results obtained in this study remains to be determined, which challenges the scientific validity and reproducibility of the study. In terms of writing, the manuscript generally resembles a first draft and needs to be rewritten for the introduction and discussion sections. Therefore, I do not recommend publication at this stage. See below for more specific comments.

1.The authors are invited to reorganize the abstract to reduce background statements and increase the presentation of important results, highlighting the innovation and the significance of this study.
2.Refine the introduction to clarify the rationale for conducting this study and its purpose and significance.
3.Line 115. NP40 lysate is usually tricky to extract the complete total protein of microsporidia. The authors should rephrase "this is the total protein of N. bombycis".
4.138 lines of "IFA" first appeared in the manuscript should provide the full name.
5.141 lines. The authors should provide the dose for 3 μL of siRNA used in the study.
6.The predicted molecular size of Ycf 1 is 34.43 kDa. However, the Western blotting in Figures 2 and 3 showed that the recombinant protein expressed by E. coli and the protein from N. bombycis was 52 kDa. I am puzzled whether the E. coli expression system undergoes phosphorylation or glycosylation modifications of proteins as the authors claim? Considering this, I would challenge the authors' statement that they are studying Ycf 1, casting doubt on the validity of the subcellular localization results of the Ycf 1 protein. Please provide compelling evidence, a plausible explanation, and an appropriate discussion for the significant difference in molecular weight. Extracting the total protein (including spore protein) with strongly denatured lysates and then analyzing by Western blotting and using mass spectrometry to identify hybridization-positive bands is recommended.
7.The "merge" in row B of Figure 4 does not seem to result from objective superposition. Moreover, besides the two ends of the meront marked by the authors, there are many other places where you can see green fluorescent signals. All of these may significantly reduce the credibility of the data. A more convincing result is recommended (transmission electron microscopy observations can be one of the alternatives).
8.In Figure 5 and Figure 6, are the anti-Nbactin and anti-SWP30 the monoclonal antibody prepared by the authors, and please describe its source in Materials & Methods. A co-localization investigation of Ycf1 and Nbactin protein in mature spores is recommended.
9.Figure 7 should provide data for a nonsense fragment negative control. More importantly, the author should analyze statistically not only between treatments but also along the time axis. Please provide a supplementary (or additional) figure with the control data and the RNAi data shown separately for the readers to evaluate the effect and reliability of the RNAi treatment during the period of the experiment. In addition, please explain why Ycf1 expression was reduced at 48 h compared to 24 h and why RNAi interference had no significant effect.
10.Figure 8. Likewise, the authors should show as a supplementary figure with the control data and the RNAi data shown separately for the readers to evaluate the effect and reliability of the RNAi treatment during the period of the experiment and whether the proliferation of spores within the host cells follows a defined pattern reported in the previous study.
11.The authors are invited to explain and discuss why RNAi had a significant effect at 96 h, while the spore proliferation was not inhibited (Figure 7), and why there was no significant effect of RNAi at 120 h but spore proliferation was significantly inhibited (Figure 8).
12.From the discussion section, the manuscript seems to be still like a first draft, without sound, in-depth and more meaningful analysis and discussion. In addition to what the reviewers suggested in the previous points of comments, the authors were invited to provide a full discussion based on the purpose and significance of this study, the biological function of Ycf1, and the other valuable results and the limitations and shortcomings of the study.
13.Manuscript has serious grammar problems. Please check it throughout.

---

## Round 0.2 · accepted · Accept

Dear Authors,

I am pleased to inform you that your article, "Identification and subcellular localization analysis of membrane protein Ycf 1 in the microsporidian Nosema bombycis", has now been accepted for publication in PeerJ. Congratulations!

Thank you for your submission and we hope you will continue to support PeerJ.

Best,

Xiaotian Tang
Academic Editor, PeerJ
xiaotian.tang@yale.edu